# Roles of Virion-Incorporated CD162 (PSGL-1), CD43, and CD44 in HIV-1 Infection of T Cells

**DOI:** 10.3390/v13101935

**Published:** 2021-09-26

**Authors:** Tomoyuki Murakami, Akira Ono

**Affiliations:** Department of Microbiology & Immunology, University of Michigan Medical School, Ann Arbor, MI 48109-0620, USA; akiraono@umich.edu

**Keywords:** transmembrane proteins, virion incorporation, trans-infection, virus attachment, CD44, PSGL-1, CD43

## Abstract

Nascent HIV-1 particles incorporate the viral envelope glycoprotein and multiple host transmembrane proteins during assembly at the plasma membrane. At least some of these host transmembrane proteins on the surface of virions are reported as pro-viral factors that enhance virus attachment to target cells or facilitate trans-infection of CD4^+^ T cells via interactions with non-T cells. In addition to the pro-viral factors, anti-viral transmembrane proteins are incorporated into progeny virions. These virion-incorporated transmembrane proteins inhibit HIV-1 entry at the point of attachment and fusion. In infected polarized CD4^+^ T cells, HIV-1 Gag localizes to a rear-end protrusion known as the uropod. Regardless of cell polarization, Gag colocalizes with and promotes the virion incorporation of a subset of uropod-directed host transmembrane proteins, including CD162, CD43, and CD44. Until recently, the functions of these virion-incorporated proteins had not been clear. Here, we review the recent findings about the roles played by virion-incorporated CD162, CD43, and CD44 in HIV-1 spread to CD4^+^ T cells.

## 1. Introduction

In addition to the envelope glycoprotein (Env), multiple host transmembrane proteins are incorporated into nascent virions during the assembly process [1]. Some of these virion-incorporated transmembrane proteins are known to facilitate HIV-1 infection. For example, virion-incorporated ICAM-1 promotes virus attachment to target cells via interactions with LFA-1 on the surface of CD4^+^ T cells [1,2,3,4]. Integrin α4β7 on the surface of virions may promote trans-infection. Virion-associated integrin α4β7 retains the ability to bind to its receptor MAdCAM-1, which is known to be present on high endothelial venules of Peyer’s patches and other gut lymphoid tissues, and therefore, the HIV-1 bound to MAdCAM-1 expressing cells in these locations may infect target CD4^+^ T cells upon cell–cell contact [1,5,6]. Anti-viral transmembrane proteins are also incorporated into progeny virions. SERINC3 and SERINC5 restrict HIV-1 fusion when these proteins are incorporated into virions [7,8,9,10]. Virion-incorporated IFITM proteins also suppress the fusion of HIV-1 [10,11,12,13,14]. Therefore, virus-incorporated host transmembrane proteins can have broad impacts, either positive or negative, on HIV-1 spread. Since the most common target cell of HIV-1 in vivo is CD4^+^ T cells, the understanding of roles played by host transmembrane proteins expressed and incorporated into virus particles in CD4^+^ T cells could contribute to the development of anti-viral strategies.

Previous studies showed that Gag localizes to uropods, which are rear-end protrusions of polarized leukocytes [15,16,17]. Gag colocalizes with a subset of uropod-directed transmembrane proteins, namely, CD162, also known as a P-selectin glycoprotein ligand-1 (PSGL-1); CD43; and CD44, in polarized T cells, and even in non-polarized cells these proteins cocluster with Gag multimers on the cell surface and are incorporated into virus particles. Therefore, it is likely that these uropod proteins are incorporated into progeny virions specifically rather than passively [5,18,19,20,21,22,23,24,25,26,27]. These three uropod-directed proteins have common features in leukocytes, localizing to the uropod in a manner that is dependent on core 1-derived O-glycans [28] and supporting the rolling of leukocytes, including T cells [29]. Interestingly, the impact of these proteins on the spread of HIV-1 between T cells is quite varied when they are incorporated into virions; PSGL-1 and CD43 inhibit HIV-1 infection, whereas CD44 enhances the spread of HIV-1 via trans-infection. In this review, we will highlight recent findings about the mechanism(s) by which HIV-1 incorporates PSGL-1, CD43, and CD44 into virions and the role of these virion-incorporated host transmembrane proteins in T cell infection with HIV-1.

## 2. Mechanism(s) That Promote the Virion Incorporation of PSGL-1, CD43, and CD44

Previous reports, including our studies, found that Gag is localized to uropods in polarized T cells [15,16,17]. In addition, antibody copatching assays [16,17] and super-resolution localization microscopy experiments [30] revealed that three uropod-directed transmembrane proteins, PSGL-1, CD43, and CD44, but not other uropod-directed proteins, ICAM-1 and ICAM-3, associate with Gag at the plasma membrane in non-polarized T cells and non-T cells. The association of PSGL-1 and CD43 with Gag requires the MA domain of Gag, especially the highly basic region (HBR). In addition, polybasic sequences at the juxtamembrane region of the PSGL-1, CD43, and CD44 cytoplasmic tails promote the association between these proteins and Gag at the surface of cells [30]. Therefore, it is highly possible that an acidic entity mediates the interactions of these uropod-directed proteins with Gag through electrostatic interactions. This acidic entity could be a cluster of acidic lipids in the inner leaflet of the plasma membrane, such as phosphatidylinositol-(4,5)-bisphosphate (PI(4,5)P_2_) or phosphatidylserine (PS). Gag can recruit acidic lipids to the virus assembly site via MA [31,32,33], and the MA HBR binds to both PI(4,5)P_2_ [34,35,36,37,38,39,40,41,42,43,44,45,46,47,48] and PS [49,50,51]. In addition, Gag multimerization facilitates PI(4,5)P_2_ clustering [32,52]. On the side of the uropod-directed proteins, although associations of PSGL-1 and CD43 with PI(4,5)P2 have not been reported, CD44 interacts with PI(4,5)P_2_ because PI(4,5)P_2_ is known to promote CD44 self-clustering [53]. These observations support the possibility that a cluster of PI(4,5)P_2_ mediates the association between the MA HBR and CD44. In addition to acidic lipids, polynucleotides including tRNAs are another candidate that could link Gag and PSGL-1, CD43, and/or CD44 because the MA HBR also interacts with RNA [41,47,48,54,55,56,57,58,59,60,61]. Identifying the acidic entity mediating the association of Gag and these uropod-directed proteins warrants future investigation.

## 3. CD44 Promotes Fibroblastic Reticular Cell-Mediated Trans-Infection

CD44 is a type-1 transmembrane protein expressed in various types of cells, including macrophages [62] and CD4^+^ T cells [63]. CD44 is one of major receptors for an extracellular matrix polysaccharide, hyaluronic acids (HAs)/hyaluronan [64]. The glycosylation and splicing patterns of CD44 are different between cell types [65,66], and the difference in glycosylation and splicing patterns affect the ability of CD44 to bind to HA [67,68]. Binding of HA to CD44 activates various signaling pathways related to cell proliferation, cell–cell adhesion, and cell migration [69]. Binding of CD44, expressed by T cells, to HA mediates cell rolling and promotes the recruitment of the cells into the inflamed peritoneal cavity [70,71,72,73]. In addition, CD44 is a ligand of E-selectin, and interactions between CD44 and E-selectin play a role in the recruitment of inflammatory T cells to inflamed tissues [74]. The virion incorporation of CD44 was first identified in the 1990s by independent research groups [19,20] and was validated by subsequent studies [23,75,76]. However, until recently, the role of virion-incorporated CD44 in the spread of HIV-1 remained unclear. We demonstrated that virion-incorporated CD44 is necessary for the trans-infection of HIV-1 mediated, by a secondary lymphoid organ (SLO) fibroblastic reticular cell (FRC) (Figure 1) [24]. The FRC is a type of stromal cell found in T cell zones [77] and B cell follicles [78] of SLOs and forms a sponge-like network. This network interacts with T cells constantly [77]. In trans-infection, cells that are not susceptible to infection capture virus particles and transmit them to virus-susceptible cells. In the case of HIV-1, mature and immature dendritic cells (DCs), B cells, and subcapsular sinus macrophages are known to capture the virus and mediate trans-infection [79,80,81,82]. These cells transfer captured HIV-1 to target CD4^+^ T cells that come into contact with them, which leads to virus transmission that is more efficient than infection by cell-free virus. Mature DCs [83,84] and macrophages [82,85] capture HIV-1 particles via interactions between CD169 on the surface of these cells and a virion-incorporated glycosphingolipid, GM3. DC-SIGN on the surface of B cells and immature DCs mediates virus capture [79,80,86] through binding to Env glycans but is dispensable for immature DC-mediated trans-infection [87]. Virus capture mediated by FRCs is inhibited by the treatment of either FRCs or HIV-1 with an anti-CD44 antibody that blocks interactions between CD44 and HA. Notably, the treatment of FRCs with hyaluronidase, which degrades HA, does not inhibit but rather enhances virus capture. In contrast, hyaluronidase treatment of HIV-1 particles prevents virus capture. Therefore, it is likely that HA bound to virion-incorporated CD44 interacts with unoccupied CD44 expressed on the surface of FRCs during FRC-mediated virus capture (Figure 1) [24].

Since interactions between CD44 and HA play a role in the internalization of HA through endocytosis [88], it is possible that HIV-1 captured by FRCs is also internalized. When virus-capturing FRCs were treated with hyaluronidase, which removes HIV-1 particles on the surface of FRCs, we observed that the FRCs can still mediate trans-infection to some extent [24]. Therefore, two populations of FRC-captured HIV-1, i.e., virions on the surface of FRCs and internalized virions, are transmittable. Similar to FRCs, immature and mature DCs likely mediate trans-infection of both surface-bound and internalized HIV-1 particles [86,87,89,90,91,92]. A portion of the HIV-1 captured by mature DCs and macrophages localizes to the CD169^+^ virus-containing compartments (VCCs) for trans-infection [85,91]. VCCs are deep and convoluted invaginations of the plasma membrane, some of which are connected to the extracellular space via narrow conduits [93,94,95]. HIV-1 in the CD169^+^ VCCs likely escapes from detection by neutralizing antibodies [91]. Therefore, it is possible that HIV-1 internalized by FRCs also escapes from neutralizing antibodies. It remains to be determined where FRC-captured HIV-1 is sequestered after virus internalization and whether sequestered HIV-1 is resistant to neutralizing antibodies.

Since the major target of HIV-1 in vivo is memory CD4^+^ T cells, which express CD44 endogenously [96], CD44 is present on the surface of both target cells and HIV-1-producing cells in vivo. Therefore, CD44 is likely incorporated into HIV-1 in vivo. In support of this notion, an anti-CD44 antibody and CD44 microbeads interact with viruses derived from the plasma of HIV-1 patients [22,76,97]. Interestingly, when HIV-1 is produced from CD44-expressing cells including PBMCs and hence incorporates CD44, cell-free infection of this HIV-1 is prevented by HA on CD44-expressing target CD4^+^ T cells at the step of virus attachment [98]. Therefore, the spread of HIV-1 among CD4^+^ T cells in vivo in the absence of FRCs might be inefficient due to the presence of HA-bound CD44 on the surface of both virus particles and target cells. However, in the presence of FRCs, which can capture HIV-1 through HA-mediated interactions between CD44 on their surface and virion-incorporated CD44, unlike CD4^+^ T cells, HIV-1 dissemination may be efficiently mediated though trans-infection in SLOs.

In addition to FRCs, mucosal fibroblasts also mediate trans-infection [99,100]. In this process, virion-incorporated CD44 and HA bound to CD44 are unlikely to contribute to virus capture because mucosal fibroblasts capture HIV-1 even when HIV-1 is produced from 293T cells that do not express CD44. Furthermore, the knockout of HA synthase 2 in mucosal fibroblasts does not affect virus capture efficiency [100], suggesting that HA on the surface of mucosal fibroblasts does not play a major role in trans-infection. Therefore, the molecular mechanisms of mucosal fibroblast- and FRC-mediated trans-infection are distinct. Since two types of fibroblastic cells isolated from different organs mediate trans-infection, other fibroblastic cells residing in different organs also may enhance the spread of HIV-1 through trans-infection.

## 4. PSGL-1 and CD43 Inhibit Virus Attachment to Target Cells

Individual host cells express a unique range of anti-viral proteins, including restriction factors. The restriction factors include several host transmembrane proteins, such as tetherin and SERINCs [101,102]. PSGL-1 and CD43 are also suggested to be restriction factors, based on their evolutionary genetic signatures, their anti-viral functions, and their susceptibility to downregulation induced upon HIV-1 infection [25,103,104,105]. PSGL-1 and CD43 are primarily expressed on the surface of lymphocytes and are known to be ligands for selectin family proteins. The interactions of PSGL-1 and CD43 with selectin proteins mediate the tethering and rolling of lymphocytes to promote cell migration into inflamed sites [29]. In addition to selectin proteins, PSGL-1 interacts with chemokines, such as CCL19, CCL21, and CCL27, to promote the recruitment of specific subsets of leukocytes into inflamed tissue or SLOs [106,107]. The expression of either PSGL-1 or CD43 in virus-producing cells restricts the infectivity of progeny virions. In addition to the uropod-directed localization pattern and the inhibitory effect on virion infectivity, PSGL-1 and CD43 share a structural feature, i.e., highly glycosylated and extended extracellular domains. These extracellular domains are estimated to be 45–50 nm long and are reported to prevent cell–cell interactions. Recently, Fu et al. and our group discovered that virion-incorporated PSGL-1 and CD43 diminish HIV-1 infection through the inhibition of virus attachment to target cells (Figure 2) [26,27]. This inhibition of virus attachment was observed regardless of whether Env is present on the virus particles or whether the virus is pseudotyped with VSV-G. Furthermore, we found that PSGL-1 and CD43 on the surface of virions also attenuated virus capture by FRCs [27]. The truncation of the extracellular domain of PSGL-1 abolishes the inhibitory effect of PSGL-1 on HIV-1 infectivity. These observations support a hypothesis that PSGL-1 and CD43 physically block virus–cell binding via extended extracellular domains, regardless of molecules mediating virus–cell binding when they are incorporated into progeny virions. Consistent with this hypothesis, the extracellular domains of PSGL-1 and CD43 are longer than the combined lengths of extracellular domains of known receptor–ligand pairs that mediate HIV–cell binding. Furthermore, cryo-electron tomography revealed that the extended pre-hairpin intermediate of Env observed between a target cell membrane and cell-attached HIV-1 in the presence of fusion inhibitors is 15.6 ± 2.8 nm [108], i.e., ~3 fold shorter than the lengths of the extracellular domains of PSGL-1 and CD43. More recently, it has been reported that mucins and mucin-like proteins, which include not only PSGL-1 and CD43 but also CD164, PODXL1, PODXL2, CD34, TMEM123, and MUC1, that have elongated extracellular domains inhibit HIV-1 attachment to target cells when these proteins are overexpressed in virus-producing cells [109]. Since these proteins abolish HIV-1 infectivity and share structural features, these proteins were named the surface-hinged, rigidly-extended killer (SHREK) family of proteins. This study provides additional support to the possible anti-viral mechanism by which PSGL-1 and CD43 sterically hinder HIV-1 attachment to target cells through their elongated extracellular domains. SHREK family proteins reduce the infectivity not only of HIV-1 but also of other enveloped viruses, such as influenza A virus (IAV) and severe acute respiratory syndrome coronavirus (SARS-CoV) and SARS-CoV-2 [26,109,110]. In addition, PSGL-1 reduces the infectivity of murine leukemia virus (MLV) [26]. Therefore, both PSGL-1 and CD43 are likely to be broad-spectrum anti-viral proteins. PSGL-1 inhibits the incorporation of Spike proteins into SARS-CoV and SARS-CoV-2 pseudovirions and the entry of virus-like particles containing SARS-CoV and SARS-CoV-2 Spike proteins [110]. The anti-viral mechanisms by which PSGL-1 and CD43 suppress IAV and/or MLV infection need to be determined.

It has been reported that PSGL-1 expression in virus-producing 293T and Jurkat cells diminishes Env incorporation into nascent virions [26,111]. In addition, PSGL-1 in virus-producing cells attenuates actin depolymerization in virions when HIV-1 is produced from 293T cells [111]. These effects of PSGL-1, which may represent additional mechanisms for its anti-viral effects, have not been studied using HIV-1 produced from primary CD4^+^ T cells. Furthermore, a recent study showed that when HIV-1 is produced from PBMCs, gp120 is incorporated into virions despite the presence of PSGL-1 on the surface of virions [76]. Therefore, further studies are needed to determine whether PSGL-1 on the surface of primary CD4^+^ T cells affects the Env content on the surface of virions and the status of actin in virions. PSGL-1 expressed in target cells was also shown to inhibit HIV-1 infection in a study that used primary CD4^+^ T cells and Jurkat cells as the target cells [25]. PSGL-1 in target cells is suggested to stabilize F-actin and restricts reverse transcription [111]. However, another group did not observe any restriction by PSGL-1 expressed in target cells [26]. This discrepancy could be due to the difference in the methods that were used in these studies, i.e., transient expression, knockdown, and knockout [25,111] versus ectopic stable expression [26] of PSGL-1. Therefore, it remains to be determined whether PSGL-1 expressed by target cells inhibits reverse transcription in physiological conditions, and if it does, whether the proposed stabilization of actin accounts for the inhibitory effect of PSGL-1.

To counteract host defense mechanisms and to achieve efficient HIV-1 spread/infection, HIV-1 encodes accessary proteins and downregulates host proteins, including restriction factors, via proteasomal and/or endosomal degradation [101,102]. HIV-1 infection reduces the expression of both PSGL-1 and CD43 on the surface of cells [25,104]. Although both Vpu and Nef downregulate these proteins [25,26], the single expression of each protein shows only a moderate reduction in the surface expression levels of PSGL-1 and CD43 compared to major targets of these proteins, such as tetherin and CD4 [103]. In a recent study, our group discovered that Gag also contributes to the downregulation of PSGL-1 and CD43 on the surface of infected cells [27]. Specifically, PSGL-1 downregulation requires Gag membrane binding, an MA-dependent association between PSGL-1 and Gag at the plasma membrane, and a p6-dependent efficient virus release. Therefore, PSGL-1 expression on the surface of infected cells is likely reduced through virion incorporation of PSGL-1 and subsequent virus release. This mode of PSGL-1 downregulation would appear to impose a disadvantage in infectivity upon a population of HIV-1 that is produced before cellular PSGL-1 levels are reduced and which incorporates a larger amount of PSGL-1 than HIV-1 released after PSGL-1 reduction. However, it is conceivable that PSGL-1 may not only suppress the spread of HIV-1 but also promote it, at least under certain conditions. Since PSGL-1 is a ligand of selectin proteins, this interaction might enhance the spread of HIV-1 via trans-infection mediated by P-selectin-expressing cells, such as endothelial cells. Consistent with this possibility, a very recent study showed that P-selectin captures HIV-1 that contains PSGL-1 and that the captured viruses could be efficiently transmitted to target CD4^+^ T cells [76]. Therefore, HIV-1 containing PSGL-1 might infect CD4^+^ T cells via trans-infection.

## 5. Conclusions

HIV-1 incorporates multiple host transmembrane proteins. These virion-incorporated proteins can affect the spread of HIV-1 positively, negatively, or both. Virion-incorporated CD44 facilitates the FRC-mediated trans-infection of HIV-1 via interactions with HA in tissue culture [24]. A majority of the spread of HIV-1 is likely to take place at SLOs [112,113], in which subcapsular sinus CD169^+^ macrophages and perhaps follicular dendritic cells mediate the trans-infection of HIV-1 [82,113,114,115,116]. FRCs can mediate trans-infection in a 3D culture system, wherein FRCs form 3D networks that resemble FRC networks in LNs. Furthermore, CD44-containing HIV-1 was disseminated efficiently in an ex vivo human tonsil culture system compared to HIV-1 without CD44 [24]. Therefore, another mode of HIV-1 spread in SLOs could be FRC-mediated trans-infection, in which virion-incorporated CD44 serves as an essential factor. However, thus far, the question of whether FRCs mediate trans-infection has not been tested in vivo. If FRCs do mediate trans-infection in vivo, the interactions between CD44 and HA may serve as a potential target of antiretrovirals.

PSGL-1 and CD43 on the surface of virions are anti-viral factors that inhibit cell-free HIV-1 infection through the blocking of virus attachment to target cells [26,27]. In addition to cell-free infection, virion-incorporated PSGL-1 and CD43 block FRC-mediated trans-infection [27]. However, since PSGL-1 and CD43 on the surface of cells prevent cell–cell interactions as well, it would be interesting to explore the effects of these proteins on cell-to-cell transmission of HIV-1, since a population of these proteins are likely present on the surface of virus-producing cells. Cell-to-cell transmission is a much more efficient mode of HIV-1 spread than cell-free infection and is thought to promote HIV-1 dissemination in vivo [117,118,119]. Therefore, the question of whether PSGL-1 and CD43 on the surface of virus-producing donor cells affect the efficiency of cell-to-cell transmission warrants future investigation. Additional studies are also needed to determine whether virion-associated PSGL-1 facilitates trans-infection of HIV-1 in vivo.

Multiple host transmembrane proteins are identified as virion-incorporated proteins, and the molecular mechanisms of their pro-viral and anti-viral effects have been gradually determined. However, the balance between the pro-viral and anti-viral effects of these individual transmembrane proteins has yet to be understood. Furthermore, the conformation, distribution patterns, and absolute numbers of these proteins on the surface of virions and the impact of these parameters on the behaviors of Env remain to be studied. Since these pieces of information could contribute to the development of novel antiretrovirals and vaccines, further investigations into these aspects of virion-incorporated host proteins are necessary.

## Figures and Tables

**Figure 1 viruses-13-01935-f001:**
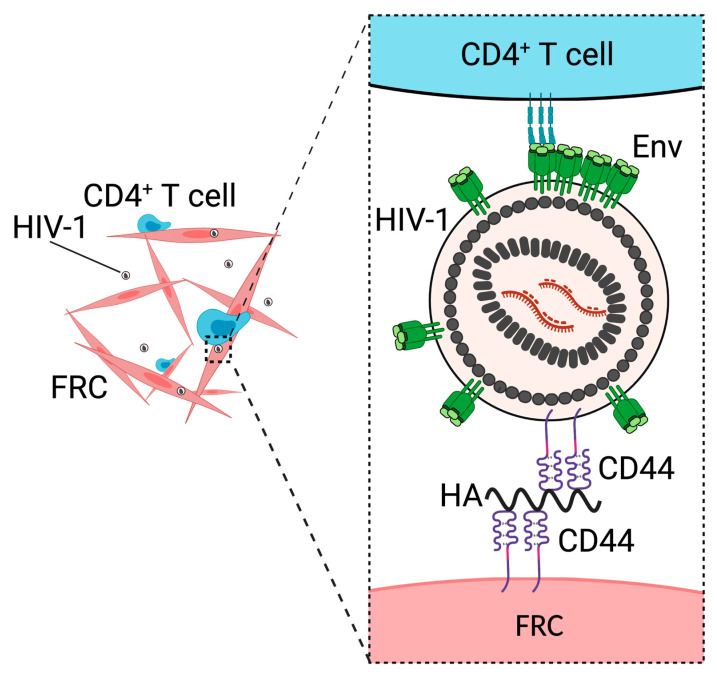
CD44 facilitates FRC-mediated trans-infection of HIV-1. FRCs capture HIV-1 particles via hyaluronan (HA)-mediated interactions between virion-associated CD44 and CD44 on the surface of FRCs. Created with BioRender.com accessed on 17 September 2021.

**Figure 2 viruses-13-01935-f002:**
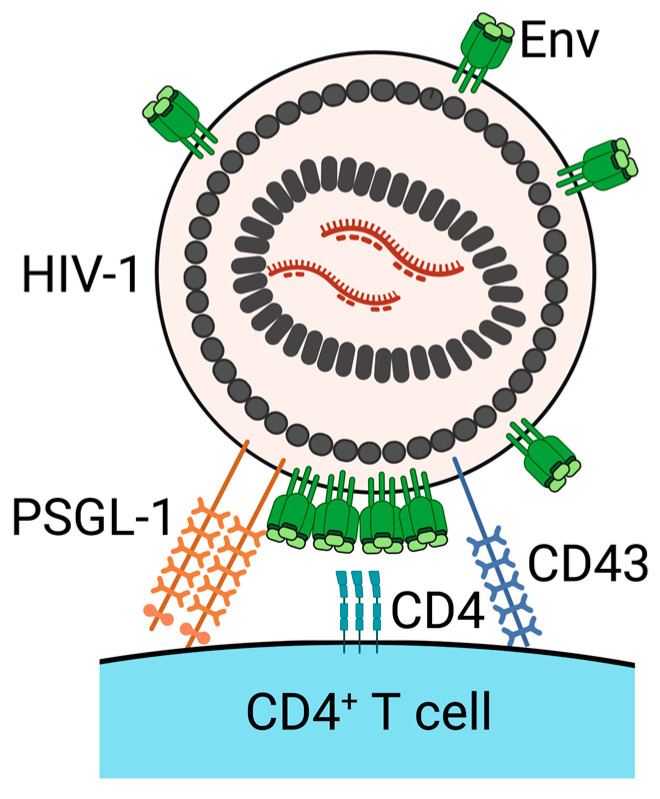
PSGL-1 and CD43 on the surface virions inhibit HIV-1 attachment to target cells. Virion-incorporated PSGL-1 and CD43 block virus attachment to target CD4^+^ T cells. Created with BioRender.com accessed on 17 September 2021.

## Data Availability

Not applicable.

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
