# Peer review of "Roles of Virion-Incorporated CD162 (PSGL-1), CD43, and CD44 in HIV-1 Infection of T Cells"

_viruses, 2021, doi:10.3390/v13101935_

Round 1

Reviewer 1 Report

  1. Minor modification of title required:
    • To keep consistent with nomenclature, title should indicate CD designation for PSGL-1, as follows: Roles of virion-incorporated CD162 (PSGL-1), CD43, and CD44 in HIV-1 infection of T cells.
    • Authors should indicate CD162 in the abstract and introduction, and then adopt PSGL-1 for the remainder of the manuscript.
  2. Line 14 – remove ‘step(s)’
  3. Line 14-15: suggested modification: '…. inhibit HIV-1 entry at the point of attachment and fusion.
  4. Line 29, add a semicolon after trans-infection.
  5. Line 32: insert ‘gut’ before lymphoid tissues.
  6. Line 34-35, merge these two sentences together.
  7. Line 38: use an alternative to ‘major natural host’ – suggestion: ‘the most common target cell’ of HIV-1.
  8. Line 44: indicate CD162 here with alternate name a P-selectin glycoprotein ligand -1 (PSGL-1), then adopt PSGL-1 from here onwards.
  9. Line 57: remove ‘s’ from promotes
  10. Line 73: clarify ‘at least’ – alternate word choice
  11. Line 77: should say and/or CD44
  12. Line 86-87: The claim of being known for nearly 30 years is not supported by these references (refs 22 and 65 are from 2006 and 2009, respectively), and will fall out of context over time. Better to say it was first identified in the 1990’s by independent research groups and validated by subsequent studies (refs 22 and 65). You can also add ref 97 here where Burnie et al showed CD44 on viruses in patient plasmas.
  13. Line 90: remove ‘s’ from cells
  14. Figure 1: HIV-1 labels on left and right side of figure are illegible (overlapping text is observed)– suggested to use arial or similar font.. The graphic for zoomed out HIV particles (on the left side) is difficult to see (they appear as small circles?) and should be updated to appear as a more stylistic representation of a virion. Recommend to use a scientific illustrator software, like BioRender.
  15. Line 103-105 is confusing. Please clarify the significance.
  16. Line 112: isn’t it inherently localized to the PM or are you alluding to internalized virus? Please clarify here.
  17. Line 114-115: these two sentences should be merged together.
  18. Line 124: should be ‘resistant’
  19. Line 126: through steric hindrance? Explain?
  20. Lines 125 – 132 is somewhat confusing. Please explain clarify.
  21. Line 135 – ref 97 also used anti-CD44 armed beads. Consider including here.
  22. Section 4. (line 151) – opening sentence should be improved. ‘Individual host cells express a unique range of anti-viral proteins, including host restriction factors’
  23. Line 165 – add ‘d’ to attenuate – should be attenuated
  24. Line 165 insert ‘the’ before extracellular domain
  25. Line 176: move ‘such as CD164 …. And MUC1’ to right after ‘mucin like proteins that have elongated ECDs’ . Remove ‘an’ before elongated and remove ‘like PSGL-1 and CD43’.
  26. Fig 2: text for HIV-1 is illegible. Update the font, as suggested for Fig 1.
  27. Line 195-196: remove ‘and that amount of virion-incorporated PSGL-1 do not inversely correlate with gp120…. of virus particles’.
  28. Line 220-221: Reword this sentence for clarity.
  29. Line 224: instead of the + superscript symbol indicate ‘P-selectin-expressing cells’
  30. Line 225: insert ‘a’ before ‘recent study’
  31. Line 226: insert ‘could be’ after ‘captured viruses’
  32. Line 226/227: should be ‘transmitted’ not ‘transmit’
  33. Line 227: remove ‘a higher amount of’ – this is arbitrary
  34. Line 232: insert ‘can’ before ‘affect HIV-1 spread’
  35. Line 233-234: merge these sentences together.
  36. Line 250: remove ‘outside of assembling or nascent virions’
  37. Line 254: update to ‘Additional studies are also needed to determine’
  38. Insert a closing statement (a sentence or two) that draws all of these emerging mechanisms together and comments on the direction of the field.

Author Response

Reviewer #1 (Remarks to the Author):

  1. Minor modification of title required:

To keep consistent with nomenclature, title should indicate CD designation for PSGL-1, as follows: Roles of virion-incorporated CD162 (PSGL-1), CD43, and CD44 in HIV-1 infection of T cells. Authors should indicate CD162 in the abstract and introduction, and then adopt PSGL-1 for the remainder of the manuscript.

[Response] We appreciate the reviewer for the helpful suggestions. As suggested by the reviewer, we changed the title and PSGL-1 to CD162 in the abstract and introduction.

  1. Line 14 – remove ‘step(s)’

[Response] We removed it.

  1. Line 14-15: suggested modification: '…. inhibit HIV-1 entry at the point of attachment and fusion.

[Response] We modified the phrasing as suggested.

  1. Line 29, add a semicolon after trans-infection.

[Response] If we add a semicolon, the sentence would become very long. So, we chose to keep the original sentence structure.

  1. Line 32: insert ‘gut’ before lymphoid tissues.

[Response] We inserted “gut”.

  1. Line 34-35, merge these two sentences together.

[Response] The original sentence in line 34 introduce the concept of virion incorporation of antiviral transmembrane proteins, which is symmetrical to the sentence in lines 26-27. Therefore, we are thinking it would be better to keep those two sentences separate. However, we would accept the merging if the editorial office regards it necessary.

  1. Line 38: use an alternative to ‘major natural host’ – suggestion: ‘the most common target cell’ of HIV-1.

[Response] We changed it as suggested.

  1. Line 44: indicate CD162 here with alternate name a P-selectin glycoprotein ligand -1 (PSGL-1), then adopt PSGL-1 from here onwards.

[Response] We changed it as suggested.

  1. Line 57: remove ‘s’ from promotes

[Response] We removed it.

  1. Line 73: clarify ‘at least’ – alternate word choice

[Response] We changed “at least” to “while associations of PSGL-1 and CD43 with PI(4,5)P2 have not been reported,” to clarify what we want to say.

  1. Line 77: should say and/or CD44

[Response] We added “/or”.

  1. Line 86-87: The claim of being known for nearly 30 years is not supported by these references (refs 22 and 65 are from 2006 and 2009, respectively), and will fall out of context over time. Better to say it was first identified in the 1990’s by independent research groups and validated by subsequent studies (refs 22 and 65). You can also add ref 97 here where Burnie et al showed CD44 on viruses in patient plasmas.

[Response] We changed the sentence as suggested by the reviewer. In addition, we cited the paper the reviewer suggested.

  1. Line 90: remove ‘s’ from cells

[Response] We think that the suggested change is grammatically incorrect. However, we would accept the change if the editorial office regards it necessary.

  1. Figure 1: HIV-1 labels on left and right side of figure are illegible (overlapping text is observed)– suggested to use arial or similar font.. The graphic for zoomed out HIV particles (on the left side) is difficult to see (they appear as small circles?) and should be updated to appear as a more stylistic representation of a virion. Recommend to use a scientific illustrator software, like BioRender.

[Response] We used BioRender to change the font and the way virions are illustrated on the left side.

  1. Line 103-105 is confusing. Please clarify the significance.

[Response] We split the sentence to two and modified both to clarify the significance.

  1. Line 112: isn’t it inherently localized to the PM or are you alluding to internalized virus? Please clarify here.

[Response] When DCs mediate trans-infection, both virus particles on the PM and inside of the VCCs (internalized virus) are transmittable. We mentioned this in the revised manuscript to address the reviewer’s question.

  1. Line 114-115: these two sentences should be merged together.

[Response] The original sentence in line114-115 just explained what are the VCCs, and the next sentence described the importance of HIV-1 internalization into the VCCs to escape from neutralizing antibodies. Thus, these sentences contain quite different information. Therefore, we did not merge the sentences.

  1. Line 124: should be ‘resistant’

[Response] We changed “resistance” to “resistant”.

  1. Line 126: through steric hindrance? Explain?

[Response] Li et al. did not determine how does HA on the surface of target cells inhibit HIV-1 binding. Since we thought this information disrupts the flow of the sentences, we did not add it to the revised manuscript.

  1. Lines 125 – 132 is somewhat confusing. Please explain clarify.

[Response] We removed “However, despite that …over autologous HA.” and added “Since HA is … suppress virus binding.” to clarify the point of the sentences.

  1. Line 135 – ref 97 also used anti-CD44 armed beads. Consider including here.

[Response] We cited the paper.

  1. Section 4. (line 151) – opening sentence should be improved. ‘Individual host cells express a unique range of anti-viral proteins, including host restriction factors’

[Response] We changed the sentence as suggested by the reviewer.

  1. Line 165 – add ‘d’ to attenuate – should be attenuated

[Response] We added “d” to “attenuate”.

  1. Line 165 insert ‘the’ before extracellular domain

[Response] We inserted “the”.

  1. Line 176: move ‘such as CD164 …. And MUC1’ to right after ‘mucin like proteins that have elongated ECDs’ . Remove ‘an’ before elongated and remove ‘like PSGL-1 and CD43’.

[Response] We removed “an” and “like PSGL-1 and CD43”. However, since SHREK family proteins include PSGL-1 and CD43, we changed the sentence to “mucin like proteins, which include not only PSGL-1 and CD43 but also … MUC1, that have elongated…”.

  1. Fig 2: text for HIV-1 is illegible. Update the font, as suggested for Fig 1.

[Response] We changed the font to clarify the labels.

  1. Line 195-196: remove ‘and that amount of virion-incorporated PSGL-1 do not inversely correlate with gp120…. of virus particles’.

[Response] We removed this part as suggested by the reviewer.

  1. Line 220-221: Reword this sentence for clarity.

[Response] We reworded the sentence “upon virus particles… levels are reduced.” to “upon a population of…after PSGL-1 reduction.” to clarify the point of the sentence.

  1. Line 224: instead of the + superscript symbol indicate ‘P-selectin-expressing cells’

[Response] We changed “+” to “-expressing”.

  1. Line 225: insert ‘a’ before ‘recent study’

[Response] We inserted “a”.

  1. Line 226: insert ‘could be’ after ‘captured viruses’

[Response] We inserted “could be” as suggested by the reviewer.

  1. Line 226/227: should be ‘transmitted’ not ‘transmit’

[Response] We changed “transmit” to “transmitted”.

  1. Line 227: remove ‘a higher amount of’ – this is arbitrary

[Response] We removed “a higher amount of”.

  1. Line 232: insert ‘can’ before ‘affect HIV-1 spread’

[Response] We inserted “can” as suggested by the reviewer.

  1. Line 233-234: merge these sentences together.

[Response] We merged the sentences as suggested by the reviewer .

  1. Line 250: remove ‘outside of assembling or nascent virions’

[Response] We removed “outside of assembling or nascent virions”.

  1. Line 254: update to ‘Additional studies are also needed to determine’

[Response] We updated the sentence as suggested by the reviewer.

  1. Insert a closing statement (a sentence or two) that draws all of these emerging mechanisms together and comments on the direction of the field.

[Response] We inserted the closing statements as suggested by the reviewer.

Reviewer 2 Report

This is a timely review that summarizes current literature on the relatively new transmembrane host factors, PSGL-1, CD43 and CD44. These proteins  incorporate into HIV-1 particles and modulate their infectivity. The review presents an unbiased view of pertinent papers focusing on the cellular functions and the mechanisms of activity of these proteins in HIV-1 virions. This publication would be quite useful for scientists working on viral restriction factors and virus-host interactions in general. 

Author Response

Reviewer #2 (Remarks to the Author):

This is a timely review that summarizes current literature on the relatively new transmembrane host factors, PSGL-1, CD43 and CD44. These proteins  incorporate into HIV-1 particles and modulate their infectivity. The review presents an unbiased view of pertinent papers focusing on the cellular functions and the mechanisms of activity of these proteins in HIV-1 virions. This publication would be quite useful for scientists working on viral restriction factors and virus-host interactions in general. 

[Response] We appreciate the reviewer for the positive comments. As suggested by the reviewer, we checked English language and spelling.

Reviewer 3 Report

The manuscript by Murakami and Ono provides a focus and informative review on the role of PSGL-1, CD43 and CD44 in HIV-1 infection as well as insights into how these molecules are packaged into the HIV virions. Overall, the review was well written. It would have been helpful to provide additional background on PSGL-1, CD43 and CD44 and their cellular function especially in the context of immune cells. In addition, highlighting if these molecules may have a role in infection of other envelope viruses would be interesting.

One specific comment, the paragraph starting at line 112 about CD169 seemed to be a tangent and the connection to CD44 was not clear. 

Author Response

Reviewer #3 (Remarks to the Author):

The manuscript by Murakami and Ono provides a focus and informative review on the role of PSGL-1, CD43 and CD44 in HIV-1 infection as well as insights into how these molecules are packaged into the HIV virions. Overall, the review was well written. It would have been helpful to provide additional background on PSGL-1, CD43 and CD44 and their cellular function especially in the context of immune cells. In addition, highlighting if these molecules may have a role in infection of other envelope viruses would be interesting.

[Response] We appreciate the reviewer for the positive comments and helpful suggestions. We added the background information about the function of PSGL-1, CD43, and CD44, especially in immune cells. Also, we highlighted the role of PSGL-1 and CD43 in infection of other enveloped viruses.

One specific comment, the paragraph starting at line 112 about CD169 seemed to be a tangent and the connection to CD44 was not clear. 

[Response] We changed the structure of the paragraph to make the flow smooth and thereby clarified the connection to CD44. 

Round 2

Reviewer 1 Report

Authors have addressed all of my queries and significantly improved the quality of the manuscript.

This article will be a meaningful and timely contribution to the field.